Genome-wide identification of xyloglucan endotransglucosylase/hydrolase gene family members in peanut and their expression profiles during seed germination

Zhu Jieqiong 1 2
Tang Guiying 2
Xu Pingli 2
Li Guowei 1 2
Ma Changle 1
Li Pengxiang 1 2
Jiang Chunyu 1 2
http://orcid.org/0000-0001-7250-8944 Shan Lei 1 2 shlei1025@sina.com
Wan Shubo 1 2 wanshubo2016@163.com
1 College of Life Science, Shandong Normal University , Jinan , China
2 Bio-Tech Research Center, Shandong Academy of Agricultural Sciences/Shandong Provincial Key Laboratory of Crop Genetic Improvement , Jinan , China
Bonetta Dario
Electronic publication date: 2022 May 17
Publication date: 2022
Volume: 10
Electronic Location ID: e13428
Received 2021 Jan 8; Accepted 2022 Apr 21
Copyright: © 2022 Zhu et al.
Copyright year: 2022
Copyright holder: Zhu et al.
License: This is an open access article distributed under the terms of the Creative Commons Attribution License, which permits unrestricted use, distribution, reproduction and adaptation in any medium and for any purpose provided that it is properly attributed. For attribution, the original author(s), title, publication source (PeerJ) and either DOI or URL of the article must be cited.
License URL: https://creativecommons.org/licenses/by/4.0/

Keywords: Peanut (Arachis hypogaea L.), xyloglucan endotransglycosidase/hydrolase (XTH), Bioinformatics, Expression profile, Seed germination

Funding: National Key R&D Program of China 2018YFD100090 Shandong Provincial Natural Science Foundation ZR2021MC054 Department of the Science and Technology of Shandong Province 2019LZGC017, YDZX20203700001861 This work was supported by the National Key R&D Program of China (2018YFD1000906), the Shandong Provincial Natural Science Foundation (ZR2021MC054), and the programs from Department of the Science and Technology of Shandong Province (2019LZGC017, YDZX20203700001861). The funders had no role in study design, data collection and analysis, decision to publish, or preparation of the manuscript.

==============================
Seed germination marks the beginning of a new plant life cycle. Improving the germination rate of seeds and the consistency of seedling emergence in the field could improve crop yields. Many genes are involved in the regulation of seed germination. Our previous study found that some peanut XTHs (xyloglucan endotransglucosylases/hydrolases) were expressed at higher levels at the newly germinated stage. However, studies of the XTH gene family in peanut have not been reported. In this study, a total of 58 AhXTH genes were identified in the peanut genome. Phylogenetic analysis showed that these AhXTHs, along with 33 AtXTHs from Arabidopsis and 61 GmXTHs from soybean, were classified into three subgroups: the I/II, IIIA and IIIB subclades. All AhXTH genes were unevenly distributed on the 18 peanut chromosomes, with the exception of chr. 07 and 17, and they had relatively conserved exon-intron patterns, most with three to four introns. Through chromosomal distribution pattern and synteny analysis, it was found that the AhXTH family experienced many replication events, including 42 pairs of segmental duplications and 23 pairs of tandem duplications, during genome evolution. Conserved motif analysis indicated that their encoded proteins contained the conserved ExDxE domain and N-linked glycosylation sites and displayed the conserved secondary structural loops 1–3 in members of the same group. Expression profile analysis of freshly harvested seeds, dried seeds, and newly germinated seeds using transcriptome data revealed that 26 AhXTH genes, which account for 45% of the gene family, had relatively higher expression levels at the seed germination stage, implying the important roles of AhXTHs in regulating seed germination. The results of quantitative real-time PCR also confirmed that some AhXTHs were upregulated during seed germination. The results of GUS histochemical staining showed that AhXTH4 was mainly expressed in germinated seeds and etiolated seedlings and had higher expression levels in elongated hypocotyls. AhXTH4 was also verified to play a crucial role in the cell elongation of hypocotyls during seed germination.

Introduction

The morphogenesis and growth patterns of plants are inseparable from the support of cell wall structures. The rigidity and elasticity of the cell wall control the size and shape of the cell, which plays a decisive role in various developmental processes in the plant (Hayashi & Kaida, 2011). Xyloglucan endotransglycosylases/hydrolases, as primary cell wall modification enzymes, mainly act on xyloglucan:xyloglucan chains, which are connected to cellulose microfibrils through hydrogen bonds and interact with adjacent microfibrils to form a network structure that provides the main mechanical support for cells (Baumann et al., 2007; Keegstra et al., 1973; Park & Cosgrove, 2015). Xyloglucan endotransglycosylases/hydrolases belong to the glycoside hydrolase family, GH16. The members of this gene family mainly perform two different biochemical functions catalyzed by two kinds of enzymes: xyloglucan endohydrolase (XEH) and xyloglucan endotransglycosylase (XET). XET activity is characterized by the nonhydrolytic cleavage and re-connection of the xylan (XyG) chain, and XEH activity irreversibly cleaves the XyG chain, promoting cell wall expansion, degradation, repair, and morphogenesis (Eklof & Brumer, 2010). According to the homology of gene sequences and the catalytic activity of their coding enzymes, XET and XEH were officially named endoglucosyltransferase/hydrolase (XTH) at the 9th International Cell Wall Meeting in 2001(Baumann et al., 2007; Eklof & Brumer, 2010; Rose et al., 2002). With the development of sequencing technology and the disclosure of data, XTH family members have been identified in an increasing number of species. It has been reported that there are 33 and 29 XTH family members in Arabidopsis thaliana and rice (Oryza sativa) (Yokoyama & Nishitani, 2001; Yokoyama, Rose & Nishitani, 2004), and the genomes of some allopolyploid plants contain a larger number of XTH members, for example, there are more than 57 in wheat (Triticum aestivum) (Liu et al., 2007a), 56 in tobacco (Nicotiana tabacum) (Wang et al., 2018), and 61 in soybean (Glycine max) (Song et al., 2018). However, the XTH family members in peanut remain unknown.

Many studies have found that XTH genes play important roles in many crucial processes during plant growth and development through remodeling of the cell wall. GhXTH1 is predominantly expressed in cotton fibers and specifically controls the elongation of fibers (Lee et al., 2010). Fruit ripening and softening are closely associated with the modification of cell wall components. FvXTH6, FvXTH9, FvXTH18, and FvXTH20 were found to take part in the process of strawberry (Fragaria vesca) ripening and softening (Opazo et al., 2017; Witasari et al., 2019). Some studies have indicated that XTHs also regulate seed germination. The expression of the CaXTH1 gene in chickpea (Cicer arietinum) gradually increased with radicle protrusion, implying that this gene might be involved in the elongation of epicotyls and embryonic axes during seed germination (Hernandez-Nistal et al., 2006). The LeXET4 gene is specifically expressed in the endosperm cap of tomato and regulates the weakening of the endosperm cap prior to radicle emergence (Chen, Nonogaki & Bradford, 2002). In addition, AtXTH18 and AtXTH31 were found to promote the elongation of Arabidopsis primary roots (Osato, Yokoyama & Nishitani, 2006). The expression of some XTH genes is also regulated by hormones, such as gibberellin (GA), auxin, and brassinolide (BR) (Rachel et al., 1996; Sasidharan et al., 2014). The expression of AtXTH19 in the elongation zone of the root was induced by auxin (Osato, Yokoyama & Nishitani, 2006), and GA could enhance AtXTH21 expression in hypocotyls (Liu et al., 2007b). OsXTH8 was also upregulated by GA and played a role in internodal cell elongation (Jan et al., 2004).

Cultivated peanut (Arachis hypogaea L.) is an allotetraploid with AA and BB subgenomes, which are derived from diploid wild peanut Arachis duranensis and Arachis ipaensis, respectively (Bertioli et al., 2019). Recently, great progress has been made in sequencing the tetraploid peanut genome, and 2.54 and 2.7 GB genome sequences were obtained from SHITOUQI and Tiffrunner, respectively (Bertioli et al., 2019; Zhuang et al., 2019). This study will be helpful for the identification and analysis of functional genes in peanut. In this study, 58 AhXTH genes derived from the peanut genome were identified and analyzed. The gene structure, evolutionary relationship, chromosomal location, and conserved domains of these AhXTH family members were investigated. In China, peanut is the important oil and protein resource, and its production area has been expanding in recent years (Zhuang et al., 2019). Maintaining the moderate dormancy of peanut seeds could guarantee pod yields by improving the germination rate and consistency of seedling emergence in the field. Previous studies have shown that genes such as EXPANSINs and XTHs are expressed at higher levels during germination (Xu et al., 2020). To clarify the functions of some AhXTHs in seed germination, the expression profiles of 58 AhXTH genes at three different stages, freshly harvested seeds (FS), dried seeds (DS) and newly germinated seeds (GS), were analyzed according to RNA-Seq data (Bioproject accession: PRJNA545858), and the transcription levels of some genes at the above stages were also explored by qRT–PCR. Furthermore, the function of AhXTH4 during the process of seed germination was verified by functional complementation in an orthologous gene mutant in Arabidopsis. These results provide a scientific basis for the functional characterization of AhXTH genes in the future.

Materials and Methods

Identification of AhXTH family members and their physicochemical properties

Two methods were used to search for the target proteins. First, conservative domain files containing PF00722 (Glyco_hydro_16) and PF06955 (XET_C) were obtained from the Pfam database (http://pfam.xfam.org/), and then they were used to search for the XTH proteins from Peanutbase (https://www.peanutbase.org/) with HMMER 3.0 software. Second, peanut XTHs were retrieved using “cultivated peanut xyloglucan endoglycosidase/hydrolase” as a keyword by searching the NCBI database (https://www.ncbi.nlm.nih.gov/). Then, all obtained XTHs were confirmed by the CDD program (https://www.ncbi.nlm.nih.gov/cdd/).

The online software ProtParam (http://web.expasy.org/protparam/) was used to predict the physical and chemical properties of the gene family members, including the number of amino acids, theoretical molecular weight (MW), and isoelectric point (PI). The subcellular locations of the XTH proteins were predicted through the online software Euk-mPLoc 2.0 (http://www.csbio.sjtu.edu.cn/bioinf/euk-multi-2/) (Chou & Shen, 2010).

Chromosomal location and collinearity analysis of AhXTHs

The physical position of each XTH gene was downloaded from Peanutbase and the NCBI database, and chromosomal location mapping was performed by MapChart 2.30.

All peanut XTH sequences were compared with each other by MCScanX (Wang et al., 2012). If two sequences shared over 70% identity and covered over 70% of their sequences, then they were considered to be homologous genes and to have collinearity (Gu et al., 2002; Yang et al., 2008). Tandemly duplicated genes were defined as adjacent homologous genes on the same chromosome with no more than one intervening gene (Zhu et al., 2014). The duplication relationship of AhXTHs was drawn by Circos mapping software (Krzywinski et al., 2009) and displayed with interconnections according to the positional information on the chromosomes. The values of nonsynonymous substitutions (Ka) and synonymous substitutions (Ks) between homologous genes were calculated by TBtools software, and the Ka/Ks value was used to predict the evolutionary relationship between homologous gene pairs.

Analysis of evolutionary relationships

The gene IDs of Arabidopsis and soybean XTH family members were obtained from published articles (Song et al., 2018; Yokoyama & Nishitani, 2001), and their amino acid sequences were downloaded from the TAIR (https://www.arabidopsis.org/) and JGI (https://phytozome.jgi.doe.gov/pz/portal.html) websites. The XTH protein sequences derived from Arabidopsis thaliana, soybean and peanut were aligned by the MUSCLE algorithm tool, and the unrooted phylogenetic tree was constructed using the NJ (neighbor-joining) method with 1,000 bootstrap replicates by MEGA-X software (Kumar et al., 2018).

Gene structure and analysis of conserved structures

The gene structure map of every AhXTH was drawn on the GSDS 2.0 (Gene Structure Display Server, http://gsds.cbi.pku.edu.cn/) website (Hu et al., 2015). The secondary structure and the conserved structural elements were predicted by the online tool ESPript (http://espript.ibcp.fr/ESPript/ESPript/) (Robert & Gouet, 2014). Using TmNXG1 (PDB id: 2UWA) (Mark et al., 2009) and PttXET16-34 (PDB id: 1UN1) (Johansson et al., 2004) as the reference sequences, all AhXTH sequences were optimally aligned by ClustalW, and the common elements of XTH secondary structures were displayed in the alignment map.

Expression profiles of AhXTHs and qRT–PCR verification

The raw data for the transcriptomes of seeds at three different stages were collected from NCBI (NCBI, Bioproject accession: PRJNA545858) (Xu et al., 2020). The three seed stages were freshly harvested seeds (FS), dried seeds (DS) that has been exposed to sunshine for two weeks, and newly germinated seeds (GS) in which the radicles broke through the seed coat. The cultivated peanut Tifrunner was used as a reference genome (gnm2. J5K5, https://www.peanutbase.org/data/public/Arachis_hypogaea/), and the RNA-seq data were analyzed by Hisat2, Samtools and Cufflinks to obtain FPKM values. A heatmap displaying gene expression patterns (log2FPKM values) was drawn by TBtools software.

Total RNA from FS, DS and GS of Fenghua No. 1 (FH1) was extracted using the RNAprep Pure Plant Plus Kit (Tiangen Biotech, Beijing, China) according to the manufacturer’s protocol, and the quality and quantity of RNA were determined by agarose gel electrophoresis and ultraviolet spectrophotometry (BioPhotometer Plus. Eppendorf, Germany), respectively. First-strand cDNA was synthesized by the PrimeScript II 1st Strand cDNA Synthesis Kit (TaKaRa, Dalian, Liaoning Province, China). Real-time PCR was performed using TB Green Premix Ex Taq with a 7,500 Fast Real-Time PCR system (Applied Biosystems, Foster City, CA, USA). The ACTIN gene was used as a reference, and the PCR conditions were as follows: 95 °C for 30 s followed by 40 cycles of 95 °C for 5 s and 60 °C for 34 s. The primers (File S1) were designed and verified on NCBI, and the results were analyzed by the 2−∆∆Ct method. Three biological replicates were performed. Pearson correlation coefficient were used to check the data consistency between RNA-seq and qRT-PCR.

Analysis of cis-acting regulatory elements in promoters

The 2,000 bp 5′-upstream sequences from the start codon (ATG) of all AhXTHs were extracted by TBtools and were used for the prediction of cis-acting regulatory elements according to PlantCARE (http://bioinformatics.psb.ugent.be/webtools/plantcare/html/) (Lescot et al., 2002).

Construction of GUS expression vectors and GUS histochemical staining

The promoter fragment of the AhXTH gene was obtained by PCR. To construct the vector, the appropriate restriction sites were introduced into the primers (HindIII at the 5′ end; NcoI at the 3′ end). The PCR-amplified products were then inserted into HindIII/NcoI-digested pCAMBIA3301, replacing the cauliflower mosaic virus (CaMV) 35S promoter. A construct harboring the AhXTH promoter was established and introduced into Agrobacterium tumefaciens strain GV3101 using the freeze–thaw method. Transgenic Arabidopsis plants were generated by the floral dip method. The homozygous T2 transgenic lines with a single-copy insertion were screened by Basta resistance. All Arabidopsis plants grew in a growth room at 23 °C under a 16 h light/8 h dark photoperiod and 65% relative humidity.

GUS histochemical staining was performed as described by Jefferson, Kavanagh & Bevan (1987). The roots and leaves at the 4-leaf stage, stems at the bolting stage, inflorescence, newly germinated seeds without testa, and etiolated seedlings germinated for 24 h and 48 h in the dark from transgenic T2 lines were incubated in GUS assay buffer with 50 mM sodium phosphate (7.0), 0.5 mM K3Fe(CN)6, 0.5 mM K4Fe(CN)6·3H2O, 0.5% Triton X-100, and 1 mM X-Gluc at 37 °C overnight and then cleared with 70% ethanol. The samples were observed by stereomicroscopy.

Identification of the functions of AhXTH4

AtXTH22 is orthologous to AhXTH4 in peanut. Its T-DNA insertion mutant xth22 (CS860818) from the Arabidopsis Biological Resource Center (ABRC, https://abrc.osu.edu) was used for functional analysis. The xth22 homozygous mutant line was selected by Basta resistance and verified by the tri-prime PCR method (primer sequences in File S2). The seeds from xth22 and Col-0 were sterilized, sown on half-strength MS medium in rows, and grown at 20 °C in the dark for 2 days; the seedlings were transferred to the light for 6 h and 24 h. The pROKII-35S::AhXTH4 vector was constructed and transformed into Agrobacterium tumefaciens strain GV3101 for use in functional complementation experiments. Transgenic Arabidopsis plants were obtained by the floral dip method and selected on 1/2 MS medium supplemented with kanamycin. Three independent homozygous lines carrying the single copy gene AhXTH4 were used for further analysis. The phenotypes of Col-0, xth22 and transgenic lines at germination, etiolated seedling establishment and photomorphogenesis were observed and photographed under an anatomical lens.

Results

Identification of XTH family members in peanut

Seventy-six and sixty-five candidate sequences of AhXTHs were retrieved using two methods, among which some incomplete sequences and sequences lacking the conserved DEIDFEFLG motif were removed. Thus, a total of 58 AhXTH protein sequences were obtained and named AhXTH1 to AhXTH58 according to their location on the chromosome (coding sequences in File S3 and protein sequences in File S4).

The length of the AhXTH genes varied from 1,187 bp to 7,499 bp, and AhXTH40 and AhXTH28 were the shortest and longest genes, respectively. Their encoded proteins ranged from 156 to 361 amino acids with an average of 292 amino acids, among which the shortest and longest proteins were AhXTH31 and AhXTH16/AhXTH45, with MWs of 17.78 and 41.8 kDa, respectively (Table 1). Analysis of the physical and chemical properties also showed that the theoretical PIs of the AhXTHs varied from 4.79 to 9.48. Most of the AhXTH members contained a signal peptide, which consisted of the first 25 amino acids of the N-terminus. Subcellular localization prediction revealed that most AhXTHs were localized on the cell wall, and some were located outside of the cell (Table 1).

Table 1 General information and physicochemical properties of identified peanut XTH genes.

Gene	ID	Chr	Chr. location	Sub-family	Gene length (bp)	No. of intron	Protein length (aa)	MW (Da)	PI	Position of signal peptide	Sub-cellular localization	
AhXTH1	XP_025692889	1	9264374..9266321	I/II	1,947	3	291	32,806.18	8.17	1~24	Extracell.	
AhXTH2	Ah4LL9T6	1	23910001..23913694	I/II	3,693	5	307	35,245.96	8.48	–	Cell wall.	
AhXTH3	Ah3BDP96	1	95261662..95263282	I/II	1,620	3	321	36,163.78	6.07	–	Cell wall. Cytoplasm.	
AhXTH4	Ah5CAM0E	1	96362584..96365077	I/II	2,493	3	300	33,487.67	8.17	1~31	Cell wall. Cytoplasm.	
AhXTH5	XP_025607418.1	1	96366146..96367469	I/II	1,324	3	296	33,710.66	7.63	1~25	Extracell.	
AhXTH6	AhS987R3	1	96375153..96376501	I/II	1,348	3	285	31,944.71	5.29	1~20	Cell wall. Cytoplasm.	
AhXTH10	AhZJ180B	3	16276361..16279298	I/II	2,937	4	302	34,460.04	6.26	1~22	Cell wall.	
AhXTH11	AhBLVJ4U	3	39788735..39791721	I/II	2,986	3	272	30,679.66	7.01	1~26	Cell wall. Cytoplasm.	
AhXTH14	XP_025691724	3	142209474..142211459	I/II	1,985	3	288	32,880.23	9.21	1~29	Cell wall. Cytoplasm.	
AhXTH15	AhWK29PC	4	1847645..1849934	I/II	2,289	3	290	32,736.82	7.7	1~28	Cell wall. Cytoplasm.	
AhXTH17	Ah4H7469	5	6221278..6223456	I/II	2,178	4	292	33,623.23	8.14	1~27	Cell wall.	
AhXTH18	AhBT8HPN	5	36362863..36365920	I/II	3,057	4	296	34,394.48	5.04	1~23	Cell wall.	
AhXTH19	AhWJ9AF6	5	96779100..96781764	I/II	2,664	5	287	32,846.15	7.7	1~26	Extracell.	
AhXTH21	Ah63MW46	6	97352601..97356107	I/II	3,506	5	286	32,833.21	8.64	1~21	Cell wall. Cytoplasm.	
AhXTH22	AhT1JMF3	6	97361543..97364727	I/II	3,184	4	295	34,788.45	9.02	1~22	Cell wall. Cytoplasm.	
AhXTH24	Ah4KDK0P	8	32459673..32462027	I/II	2,354	4	295	33,451.48	5.57	1~20	Cell wall.	
AhXTH25	AhTC32UL	9	111679851..111681534	I/II	1,683	5	296	34,569.05	6.7	–	Cell wall.	
AhXTH27	XP_025626284	11	1830396..1832678	I/II	2,282	3	291	32,814.2	8.16	1~24	Extracell.	
AhXTH28	AhQH33BI	11	28044899..28052398	I/II	7,499	4	298	33,848.31	7.03	–	Cell wall.	
AhXTH29	XP_025630721	11	147786611..147788180	I/II	1,569	3	285	31,930.64	5.13	1~20	Cell wall. Cytoplasm.	
AhXTH30	AhB3UXLR	11	147790372..147792104	I/II	1,732	3	283	32,097.33	9.05	1~24	Cell wall. Cytoplasm.	
AhXTH31	AhJFZB9X	11	147793950..147795665	I/II	1,715	4	156	17,785.92	4.79	–	Cell wall. Cytoplasm.	
AhXTH32	AhYY6L6Q	11	147801864..147803262	I/II	1,398	3	296	33,710.66	7.63	1~25	Extracell.	
AhXTH33	AhV02MI5	11	147804129..147806580	I/II	2,451	3	297	33,224.38	6.81	1~29	Cell wall. Cytoplasm.	
AhXTH34	AhBWTU0I	11	148648513..148650124	I/II	1,611	3	282	31,594.34	5.82	1~21	Cell wall. Cytoplasm.	
AhXTH35	XP_025632787	12	36960806..36962430	I/II	1,624	3	292	32,390.21	5.49	1~23	Cell wall. Cytoplasm.	
AhXTH38	AhI4AHF2	13	1087056..1089406	I/II	2,350	3	288	32,866.21	9.21	1~29	Cell wall. Cytoplasm.	
AhXTH39	XP_025643304	13	18501534..18504029	I/II	2,495	4	302	34,501.14	6.39	1~22	Cell wall.	
AhXTH41	AhVDQ89P	13	41806175..41808848	I/II	2,673	3	272	30,689.7	7.01	1~26	Cell wall. Cytoplasm.	
AhXTH44	Ah8EK9UN	14	2533475..2536240	I/II	2,765	3	280	31,602.65	8.33	1~23	Cell wall. Cytoplasm.	
AhXTH46	Ah2QGA82	15	6221278..6223456	I/II	2,178	4	292	33,623.23	8.14	1~27	Cell wall.	
AhXTH47	AhGX4GBD	15	21015893..21018935	I/II	3,042	4	296	34,391.52	5.03	1~23	Cell wall.	
AhXTH48	XP_025650986	15	148997430..149000064	I/II	2,634	4	291	33,304.72	7.64	1~26	Extracell.	
AhXTH51	AhB1AFV9	16	127800516..127804081	I/II	3,565	4	277	32,042.34	8.8	1~21	Cell wall. Cytoplasm.	
AhXTH52	AhX2UPFJ	16	127809583..127812339	I/II	2,756	4	295	34,759.39	8.93	1~22	Cell wall. Cytoplasm.	
AhXTH54	Ah6NV05N	18	8787552..8789993	I/II	2,441	4	295	33,499.53	5.58	1~20	Cell wall.	
AhXTH55	AhJZ1BNU	19	15091283..15097771	I/II	6,488	4	262	30,134.86	8.56	–	Extracell.	
AhXTH56	XP_025678148	19	15121519..15123877	I/II	2,358	4	289	33,285.15	5.43	1~21	Extracell.	
AhXTH57	AhKYWB2X	19	156537033..156538442	I/II	1,409	4	295	34,398.86	7.08	1~24	Cell wall.	
AhXTH8	Ah31HZRQ	2	102299085..102304062	IIIA	4,977	7	334	38,891.24	9.48	1~22	Cell wall.	
AhXTH9	Ah0LE9N4	3	968426..972658	IIIA	4,232	4	304	34,656.06	7.1	1~29	Cell wall.	
AhXTH20	AhT3CFRI	6	41986865..41990953	IIIA	4,088	4	293	34,089.7	9.19	1~17	Cell wall.	
AhXTH23	Ah68HMAV	8	29429382..29431027	IIIA	1,645	4	300	34,146.2	5.47	1~29	Cell wall.	
AhXTH37	AhA5JVQI	12	119897010..119901987	IIIA	4,977	7	334	38,891.24	9.48	1~22	Cell wall.	
AhXTH49	Ah4SWV4Y	15	155559622..155563443	IIIA	3,821	4	281	32,631.97	9.23	1~23	Cell wall.	
AhXTH50	AhZ02ZAB	16	50226796..50230387	IIIA	3,591	4	258	29,875.77	9.19	–	Cell wall.	
AhXTH53	Ah0PHB1J	18	5146751..5148597	IIIA	1,846	4	300	34,117.2	5.65	1~29	Cell wall.	
AhXTH7	AhU5U9K5	2	97588921..97590688	IIIB	1,767	2	217	24,663.54	6.1	–	Cell membrane. Extracell.	
AhXTH12	AhE945XK	3	133223992..133227112	IIIB	3,120	4	305	34,127.01	8.5	1~24	Extracell.	
AhXTH13	AhVWTL5B	3	140540321..140543117	IIIB	2,796	4	334	37,861.63	6.42	1~23	Cell wall.	
AhXTH16	XP_025697906	5	1640387..1645019	IIIB	4,632	4	361	41,892.11	8.62	–	Cell wall.	
AhXTH26	AhK91ZD2	10	115928757..115933233	IIIB	4,476	4	336	38,156.96	6.67	1~21	Cell wall.	
AhXTH36	Ah0BZZ1B	12	114137012..114138710	IIIB	1,698	2	217	24,534.49	6.1	–	Cell membrane. Extracell.	
AhXTH42	AhN9VL7S	13	135817489..135820499	IIIB	3,010	8	306	34,201.05	8.16	1~25	Extracell.	
AhXTH43	AhMAC6YY	13	143519589..143522307	IIIB	2,718	4	334	37,806.54	6.26	1~23	Cell wall.	
AhXTH45	XP_025651253	15	1640387..1645019	IIIB	4,632	4	361	41,892.11	8.62	–	Cell wall.	
AhXTH58	AhD3GWAL	20	142725442..142730103	IIIB	4,661	4	321	36,502.97	6.67	1~21	Extracell.	
Note:

“-” indicates no position of signal peptide.

Chromosomal locations and duplicated events of AhXTHs

The mapping analysis of AhXTH genes on chromosomes showed that except for chromosome (chr.) 07 and 17, all 58 AhXTH genes were unevenly distributed on the remaining 18 chromosomes and mostly concentrated at both ends of the chromosomes. Additionally, 26 and 32 of them belonged to the A genome (chromosomes 01–10) and B genome (chromosomes 11–20) (Fig. 1), respectively. In detail, there are eight AhXTH genes on chr. 11, which held the largest number of AhXTH gene; six AhXTHs were scattered on chr. 01, 03, and 13, and five AhXTH genes were found on chr. 15. The number of genes distributed on these five chromosomes accounted for approximately 53% of the total number of genes. The remaining 13 chromosomes unequally contained 1 to 4 AhXTH genes, accounting for less than 50% of the genes. The results also showed that five gene clusters, including AhXTH3-XTH6; AhXTH29-XTH34; AhXTH21 and AhXTH22; AhXTH51 and AhXTH52; and AhXTH55 and AhXTH56, were located on chr. 01, 11, 06, 16, and 19 (Fig. 1).

Figure 1 Distribution of all peanut AhXTH genes on chromosomes.

The 58 AhXTH genes were unevenly distributed on the 18 chromosomes, with the exception of chr. 07 and 17. The location on the chromosome of each AhXTH gene was indicated on the right side of the respective chromosome. The gene names belonging to the I/II, the IIIA and the IIIB subfamily were respectively represented by black, red, and blue words. The red box represented the clusters of tandemly duplicated genes. The scale bar for chromosome length was showed at the left of all chromosomes.

The syntenic relationships among all AhXTH genes were assessed based on sequence homology and positional information. A total of 65 replication events, including 42 pairs of segmental duplications and 23 pairs of tandem duplications from five gene clusters located in an interval of less than 100 kb, were recognized; these are shown on the Circos map (Fig. 2). To explore the evolutionary relationship among AhXTH members, the synonymous (Ks), nonsynonymous (Ka) and Ka/Ks ratio values for each duplication event were calculated (Table 2). The Ka values of the segmental duplications and tandem duplications ranged from 0 to 0.7027 and from 0.1226 to 0.3342, respectively, while the Ks values ranged from 0 to 2.3858 and from 0.5394 to 2.8151, respectively. The Ka/Ks ratio of the AhXTH25/AhXTH57 segmental duplication was more than one, suggesting that this gene pair underwent positive selection, while the other pairs of segmental duplications and all gene pairs with tandem duplications had a Ka/Ks < 1 and were negatively selected during evolution. The results suggested that during the evolution and amplification of the genome, most AhXTH genes may have undergone purification selection on the codons.

Figure 2 Schematic representations of colinearity analysis of the AhXTH genes.

The synteny relationship among AhXTH genes was shown as Circos map. The red line indicates the duplicated events between gene pairs, and the different colored bars represent the different chromosome. The gene names were displayed on the outside of different chromosome.

Table 2 Duplicated events of peanut AhXTH genes.

Seq 1	Seq 2	Ks	Ka	Ka/Ks	Type of duplication	Seq 1	Seq 2	Ks	Ka	Ka/Ks	Type of duplication	
AhXTH49	AhXTH37	0	0.005	0	Segmentally	AhXTH1	AhXTH27	0.012	0.031	0.3774854	Segmentally	
AhXTH8	AhXTH49	0	0.005	0	Segmentally	AhXTH12	AhXTH42	0.013	0.033	0.3960755	Segmentally	
AhXTH43	AhXTH13	0.003	0.035	0.0738125	Segmentally	AhXTH19	AhXTH48	0.024	0.046	0.5181498	Segmentally	
AhXTH37	AhXTH50	0.054	0.602	0.08956	Segmentally	AhXTH18	AhXTH47	0.006	0.011	0.5444887	Segmentally	
AhXTH8	AhXTH50	0.054	0.602	0.08956	Segmentally	AhXTH21	AhXTH51	0.024	0.036	0.6825019	Segmentally	
AhXTH39	AhXTH10	0.003	0.026	0.1068308	Segmentally	AhXTH57	AhXTH25	0.006	0.005	1.0908225	Segmentally	
AhXTH17	AhXTH28	0.143	1.276	0.1121371	Segmentally	AhXTH46	AhXTH17	0	0	NaN	Segmentally	
AhXTH46	AhXTH28	0.143	1.276	0.1121371	Segmentally	AhXTH8	AhXTH37	0	0	NaN	Segmentally	
AhXTH58	AhXTH26	0.004	0.034	0.1178242	Segmentally	AhXTH38	AhXTH13	0.703	NaN	NaN	Segmentally	
AhXTH17	AhXTH2	0.134	1.123	0.1195105	Segmentally	AhXTH34	AhXTH29	0.181	2.224	0.0816139	Tandem	
AhXTH46	AhXTH2	0.134	1.123	0.1195105	Segmentally	AhXTH3	AhXTH6	0.18	2.131	0.0842365	Tandem	
AhXTH37	AhXTH20	0.073	0.607	0.1206398	Segmentally	AhXTH3	AhXTH4	0.246	2.815	0.0873522	Tandem	
AhXTH8	AhXTH20	0.073	0.607	0.1206398	Segmentally	AhXTH30	AhXTH29	0.132	1.389	0.0952672	Tandem	
AhXTH44	AhXTH38	0.139	1.141	0.1220149	Segmentally	AhXTH34	AhXTH33	0.252	2.578	0.0977924	Tandem	
AhXTH14	AhXTH44	0.139	1.096	0.1269737	Segmentally	AhXTH30	AhXTH31	0.174	1.715	0.1012063	Tandem	
AhXTH38	AhXTH15	0.153	1.093	0.1401732	Segmentally	AhXTH30	AhXTH34	0.18	1.748	0.1028589	Tandem	
AhXTH4	AhXTH29	0.199	1.343	0.1480976	Segmentally	AhXTH29	AhXTH33	0.201	1.542	0.1304063	Tandem	
AhXTH14	AhXTH15	0.153	1.012	0.1513226	Segmentally	AhXTH34	AhXTH31	0.23	1.655	0.1392606	Tandem	
AhXTH36	AhXTH7	0.006	0.036	0.1644488	Segmentally	AhXTH3	AhXTH5	0.313	2.227	0.1406897	Tandem	
AhXTH40	AhXTH48	0.157	0.939	0.1671528	Segmentally	AhXTH4	AhXTH6	0.197	1.376	0.1431129	Tandem	
AhXTH11	AhXTH41	0.006	0.037	0.1733513	Segmentally	AhXTH31	AhXTH29	0.134	0.853	0.1569053	Tandem	
AhXTH53	AhXTH23	0.006	0.033	0.1763009	Segmentally	AhXTH34	AhXTH32	0.324	2.042	0.1587536	Tandem	
AhXTH2	AhXTH11	0.44	2.386	0.184235	Segmentally	AhXTH31	AhXTH33	0.231	1.349	0.1712363	Tandem	
AhXTH24	AhXTH54	0.003	0.016	0.1845618	Segmentally	AhXTH5	AhXTH6	0.307	1.68	0.1830121	Tandem	
AhXTH40	AhXTH19	0.178	0.955	0.1864823	Segmentally	AhXTH29	AhXTH32	0.288	1.487	0.1939271	Tandem	
AhXTH11	AhXTH34	0.153	0.819	0.1871329	Segmentally	AhXTH30	AhXTH33	0.222	1.098	0.2023634	Tandem	
AhXTH34	AhXTH41	0.149	0.788	0.1893404	Segmentally	AhXTH31	AhXTH32	0.249	1.207	0.2065566	Tandem	
AhXTH3	AhXTH11	0.162	0.82	0.1972279	Segmentally	AhXTH30	AhXTH32	0.301	1.43	0.210577	Tandem	
AhXTH3	AhXTH34	0.003	0.015	0.1993766	Segmentally	AhXTH51	AhXTH52	0.123	0.539	0.2273727	Tandem	
AhXTH3	AhXTH41	0.158	0.79	0.1997005	Segmentally	AhXTH33	AhXTH32	0.314	1.307	0.2404674	Tandem	
AhXTH44	AhXTH15	0.017	0.082	0.209474	Segmentally	AhXTH4	AhXTH5	0.334	1.226	0.2726071	Tandem	
AhXTH9	AhXTH23	0.169	0.711	0.2376792	Segmentally	AhXTH21	AhXTH22	0.161	0.587	0.2739422	Tandem	
AhXTH9	AhXTH53	0.17	0.672	0.2527505	Segmentally							

Phylogenetic relationship of AhXTH members

A total of 152 XTH sequences derived from Arabidopsis, soybean, and peanut were aligned by ClustalW, and their evolutionary relationship was analyzed by MEGA-X software (Fig. 3). In previous studies, AtXTH members were first classified into three groups (Yokoyama & Nishitani, 2001); however, further analysis showed that the XTHs in Groups I and II have no clear demarcation, and Group III could also be divided into two branches, IIIA and IIIB (Yokoyama, Rose & Nishitani, 2004). Therefore, according to the cluster criterion in Arabidopsis, 58 AhXTH members together with 31 AtXTHs and 61 GmXTHs were divided into three subfamilies I/II, IIIA and IIIB on the unrooted phylogenetic tree, among which 40 AhXTHs belonged to subfamily I/II, and subfamilies IIIA and IIIB contained 8 and 10 AhXTH members, respectively (Fig. 3).

Figure 3 The phylogenetic relationship of XTHs derived from peanut, Arabidopsis, and soybean.

The unrooted phylogenetic tree was constructed using MEGA-X by neighbor-joining method with 1,000 bootstrap replicates. The XTHs of peanuts, Arabidopsis and soybeans were respectively shown in red square, yellow triangle, and blue circle. Total of 152 XTH members were classified into three subfamilies: I/II, IIIA and IIIB, and were differentiated by the folding lines with black, red and blue color.

Structural analysis of AhXTH genes and their encoded proteins

The gene structure of each AhXTH gene was drawn using GSDS 2.0 software (Fig. 4B). The prediction results showed that the numbers of AhXTH exons ranged from 2 to 8 across all family genes; genes containing 3–4 exons accounted for 85% of the total genes, and AhXTH42 contained the largest number of exons at 8. Additionally, the numbers and sizes of the exons were significantly different within each AhXTH subfamily. The homologous genes with the higher similarity of sequences had higher structural identity. The majority of the AhXTH genes had normal 5′UTR and 3′UTR structures, whereas AhXTH25, 31, 44, and 50 only had the 5′UTR structure, AhXTH8, 19, 37, 49, and 58 only had the 3′UTR structure, and AhXTH40, 48 and 56 lacked both a 5′UTR and 3′UTR. In addition, all AhXTHs contained the active site situated in front of their second or third exons.

Figure 4 Structure analysis of AhXTH genes and identification of conserved domain of their encoded proteins.

(A) The unrooted phylogenetic tree of 58 peanut AhXTHs; (B) The exon–intron structure of peanut AhXTH genes. The coding sequences (CDS) and the untranslated regions (UTR) were indicated by yellow boxes and blue boxes. The boxes in red color highlighted the active site of AhXTHs; (C) Multiple alignment of partial amino acid sequences of peanut AhXTHs for showing the conserved secondary structures. Amino acid sequences were aligned using PttXET16-34 (1UN1) and TmNXG1(2UWA) as the referent sequences by MEGA X, and their secondary structures were predicted using ESPript. The conserved residues were shown in grey frames, among which the identity residues were indicated by white letters in red boxes, and the similar residues by black letters in yellow boxes. The secondary structures of β sheets and α-helices were shown on the top of sequences with arrows and spirals. The conserved catalytic domain (DEIDFEFLG), and loops 1, 2 and 3 were indicated using red lines on the bottom of sequences. N-glycosylation residues were indicated as asterisks. (The alignment results of intact amino acid sequences of 58 AhXTHs was presented in Files S5 and S6)

The secondary structure of the AhXTH protein was predicted according to the structure of the endoxyloglucan endoglycosidase PttXET16-34 and endoxyloglucan enzyme TmNXG1, and the conserved domains are shown in Fig. 4C. The results showed that all AhXTH sequences contained the active site ExDxE. Previous studies indicated that the N-glycosylation site (NxT/S/Y) near the active site can be combined with N-glycans and is related to the stability of the protein (Campbell & Braam, 1998; Opazo et al., 2010). Our analysis showed that Group I/II members had consistent N-glycosylation sites (NxS/T) close to the active site, marked with an asterisk, “*” in Fig. 4C. However, in Group III, the N-linked glycosylation sites (NxY) of all AhXTHs were approximately 11 amino acids away from the active site. Almost all AhXTHs also contained three conserved loop structures, loop1, loop2 and loop3, and on average, the amino acid composition of loop3 was more conserved in all AhXTHs, with major differences manifested between loop1 and loop2 (Fig. 4C). Loop1 is a unique extension of the active site; it had the same number of amino acids in the IIIA and IIIB groups, but the amino acid composition was significantly different. In Group I/II, loop1 lacked three amino acids compared with that in the III subfamily, but its function was rarely influenced. Loop2 can regulate the ability to bind the substrate and determine endoxyloglucanase activity (Baumann et al., 2007). The AhXTHs of Groups I/II and IIIB lacked 3–4 amino acids in loop2 compared with most Group IIIA members. Loop3 was detectable in 55 AhXTH proteins except for AhXTH7, AhXTH31, and AhXTH36; this loop had the amino acid sequence D(S/N/A)WATR(D/Q)G(S/W) in I/II subfamily members in peanut, and the sequence SWATEN(D)GG in IIIA, while its structure and amino acid sequence in the IIIB members was more complicated, with no identical sequences (Files S5 and S6).

Expression profiles of AhXTHs at the FS, DS and GS stages

The expression levels of AhXTH genes at three stages of peanut maturation and germination were investigated using transcriptome data, and some of the genes were verified by qRT–PCR (Fig. 5). The heatmap analysis showed that the expression levels of almost half of the genes (25 genes: AhXTH1, 2, 7, 8, 9, 10, 12, 17, 19, 20, 21, 22, 26, 27, 28, 36, 37, 39, 46, 48, 49, 50, 53, 56, and 58) did not change at the FS, DS and GS stages, and 7 genes (AhXTH5, 23, 31, 32, 33, 55, and 57) were expressed at higher levels in FS than in DS and GS. Compared with those in FS and DS, the remaining 26 genes (AhXTH3, 4, 6, 11, 13, 14, 15, 16, 18, 24, 25, 29, 30, 34, 35, 38, 40, 41, 42, 43, 44, 45, 47, 51, 52, and 54) were expressed at the highest level in GS, among which AhXTH6, 11, 25, 41, and 44 were only expressed in GS, and the expression of AhXTH15, 24, 29, 52, and 54 was not detectable in FS and DS. It is worth noting that some highly homologous genes had different expression patterns. AhXTH4 and AhXTH25 had high expression levels in GS, while their orthologs AhXTH33 and AhXTH57 were expressed at a higher level in FS. AhXTH51 and AhXTH52 from one cluster were situated on chr. 16 and had higher expression in GS, while their homologous cluster members located on chr. 06, AhXTH21 and AhXTH22, were found to have consistent expression levels at all three stages. Similarly, AhXTH55 and its cluster member AhXTH56, as well as AhXTH23 and its ortholog AhXTH53, also displayed different expression patterns; the former was highly expressed in FS, while the latter had no expression at all three stages (Fig. 5A, File S7). In this study, the expression patterns of 12 AhXTH genes were verified by qRT–PCR analysis at the DS, FS and GS stages. The consistency analysis of the results revealed a moderate correlation between RNA-seq and qRT-PCR, which may be caused by the different batches of samples (File S8). For ten (AhXTH4, 14, 15, 16, 24, 30, 35, 38, 42 and 52) of them, their expression levels at the GS stage were confirmed to be significantly upregulated (Fig. 5B, Files S9 and S10). The qRT–PCR results also verified that AhXTH5 expression in FS was significantly higher than that in GS and DS, and its expression level was not obviously different between GS and DS. The AhXTH31 gene was expressed at the highest level in FS and the lowest level in DS (Fig. 5B).

Figure 5 The expression profile of peanut AhXTH genes at FS, DS and GS stages.

(A) Heatmap of AhXTH gene expression in FS, DS and GS was drawn by TBtools according to the RNA-Seq data (Bioproject_accession: PRJNA545858). The relative expression level of each gene at every stage was transformed by log2 of its FPKM value, and was displayed in colored box. The red and blue box respectively indicates the highest and the lowest expression level. No changes of expression levels at the three stages were shown as box in dark yellow color. (B) Analysis of the mRNA transcript levels of several AhXTHs at DS, FS and GS stages by qRT-PCR. The relative expression levels were calculated using ACTIN7 as an internal control by the 2−∆∆Ct method. Three biological replicates were performed. The significance of variant between DS and FS or GS were analyzed by one way ANOVA. The * and ** respectively represents significant difference at level of p<0.05 and p<0.01. The ns indicates no significant difference. FS: the freshly harvested seed; DS: the dried seed; GS: the newly germinated seed.

Analysis of the cis-acting elements of AhXTH promoter regions

Cis-acting elements play a crucial role in gene function, and the gene sequence itself can possibly influence the final expression level (Zhao et al., 2020). In this study, 2,000 bp promoter regions were analyzed using PlantCARE. The results showed that these regions contain not only common regulatory elements such as CAAT boxes and TATA boxes but also harbor numerous cis-acting elements involved in low temperature and drought responsiveness and responses to multiple hormones. In detail, the 5′ regulatory regions of the majority of AhXTHs (47 genes and 46 genes) have multiple abscisic acid responsive elements (ABREs) and ethylene responsive elements (EREs), and in the regulatory regions of 25, 26, 29 and 42 AhXTH genes, several GA-, auxin-, salicylic acid- and MeJA-related responsive elements exist (Fig. 6, Table 3).

Figure 6 The cis-acting elements predicted in the promoter regions of AhXTH genes.

The 2,000 bp fragment upstreamed from start codon (ATG) of every AhXTH was retrieved by TBtools, and analyzed using online software PlantCARE. The different cis-acting elements were shown as different colorful boxes.

Table 3 Analysis of cis-elements in the promoters of AhXTHs.

Element name	Motif sequence	Gene number with conserved motif	Related biological function	
ABRE	ACGTG	47	Abscisic acid responsiveness	
TGA-element/AuxRR-core	AACGAC/GGTCCAT	26/5	Auxin responsiveness	
TCA-element	CCATCTTTTT	29	Salicylic acid responsiveness	
TGACG-motif/CGTCA-motif	TGACG/CGTCA	42/42	MeJA responsiveness	
P-box/TATC-box/GARE-motif	CCTTTTG/TATCCCA
/TCTGTTG	11/8/15	Gibberellin responsiveness	
LTR	CCGAAA	19	Low-temperature responsiveness	
MBS	CAACTG	30	Drought-inducibility	
CAT-box	GCCACT	19	Meristem expression	
RY-element	CATGCATG	3	Seed-specific regulation	
MBS	CAACTG	30	Drought-inducibility	
circadian	CAAAGATATC	14	Circadian control	
ARE	AAACCA	46	Anaerobic induction	
ERE	ATTTTAAA	46	Ethylene responsiveness	
Note:

The cis-acting regulatory elements in the promoters of AhXTHs were predicted by PlantCARE (http://bioinformatics.psb.ugent.be/webtools/plantcare/html/).

Analysis of AhXTH expression patterns by the GUS expression system

To explore the temporal and spatial expression patterns of AhXTH genes, GUS expression vectors driven by AhXTH promoters were established, and the GUS expression patterns in stable transgenic lines of Arabidopsis harboring the AhXTH4 and AhXTH22 promoters (pAhXTH4:GUS and pAhXTH22:GUS) were investigated. The size of AhXTH4 and AhXTH22 promoter cloned is respectively 1,287 and 1,919 bp (File S11). The results of GUS histochemical staining in different tissues and organs showed that in the seedlings at the 4-leaf stage, weak signals were observable only in the veins of rosette leaves of pAhXTH4:GUS plants, while in pAhXTH22:GUS plants, the vascular tissues in rosette leaves or roots were stained blue. In stems and cauline leaves, stronger staining was found only in the leaf margins and trichomes of pAhXTH4:GUS plants, while it was found in stems and whole leaves of pAhXTH22:GUS plants. The flowers of both pAhXTH4:GUS and pAhXTH22:GUS plants were dyed dark blue, and almost no staining was observed in the siliques of pAhXTH4:GUS plants, while weaker GUS expression was found in pAhXTH22:GUS siliques (Fig. 7A). During seed germination and the establishment of etiolated seedlings, GUS expression analysis showed that both AhXTH4 and AhXTH22 had similar temporal-spatial expression patterns, and only AhXTH4 had a higher expression level. Except for a lack of staining in the radicle or root tips and the apical hook of pAhXTH4:GUS plants, all other tissues and organs in newly germinated seeds without testa and seedlings germinated for 24 h and 48 h in the dark displayed stronger GUS expression, especially the hypocotyls and primary roots of 48 h dark-exposed seedlings (Fig. 7B). This result implied that AhXTH4 might play a major role in hypocotyl elongation during seed germination and etiolated seedling establishment.

Figure 7 Analysis of AhXTH expression patterns by the GUS expression system.

The wild-type Arabidopsis Col-0 was used as a negative control in this experiment, and the transgenic line of Arabidopsis harboring the GUS expression construct driven by Cauliflower mosaic virus 35S promoter was used as a positive control. (A) GUS expression in different tissues and organs. a: siliques, b: flowers, c: stems and leaves, d: seedlings at the 4-leaf stage. (B) GUS expression in germinated seeds (imbibition for 12 h in dark) and etiolated seedlings germinated for 24 h and 48 h in dark.

Ectopic expression analysis of AhXTH4 in Arabidopsis plants with mutated orthologous genes

To validate the roles of AhXTH4 in promoting seed germination and etiolated seedling establishment, the phenotypes of its corresponding mutant xth22 in Arabidopsis and the transgenic lines constitutively expressing AhXTH4 in xth22 were investigated. The results showed that compared to wild-type Arabidopsis Col-0, xth22 showed a slower germination speed and shorter hypocotyl. When germinating in the dark for 2 days, the hypocotyl and radicle lengths of xth22 were much shorter than those of Col-0, while the overexpression of AhXTH4 in xth22 resulted in phenotypes similar to those of Col-0 during seed germination and etiolated seedling establishment (Fig. 8). After the etiolated seedlings were transferred to light for 6 h, both Col-0 and the mutant showed reduced bending of the apical hook and inhibited elongation of the hypocotyl; a similar phenotype was also found in Arabidopsis transgenic lines (Fig. 8A). Under light for 24 h, the seedlings of Col-0 and the transgenic lines had fully open and greenish cotyledons, while the cotyledons of xth22 were incompletely stretched (Fig. 8A); however, there was no significant difference in hypocotyl length among the wild-type and three transgenic lines (Fig. 8B). This result suggested that AhXTH4 was involved in the regulation of hypocotyl elongation during seed germination in the dark.

Figure 8 Ectopic expression analysis of AhXTH4 in Arabidopsis mutant xth22.

A. The phenotypes of xth22, Col-0, and transgenic lines at germinated for 1 and 2 days in dark, and transferred in light for 6 and 24 h. The bars in figure indicate 1mm. (B and C). The statistical graphs of hypocotyl (B) and radicle or root (C) lengths in xth22, Col-0 and transgenic lines. AtXTH22 is orthologous to AhXTH4 in peanut. Its T-DNA insertion mutant xth22 (CS860818) was used to perform functional complement experiment. OE1-3/xth22 were three transgenic homozygous lines overexpressing AhXTH4 in xth22.

Discussion

To date, the XTH family has been identified in various species, with 33 genes found in Arabidopsis (Yokoyama & Nishitani, 2001), 29 in rice (Yokoyama, Rose & Nishitani, 2004), 61 in soybean (Song et al., 2018), and 56 in tobacco (Wang et al., 2018). These genes take part in many important biological processes and play important roles in cell wall reconstruction. In our study, we identified 58 AhXTH family members in cultivated peanut, which combined with XTHs from Arabidopsis and soybean, were classified into three subfamilies: I/II, IIIA, and IIIB. The gene structure and conserved motifs of these genes, as well as their chromosomal location and collinearity relationship were analyzed in detail. The expression profiles of AhXTHs were also investigated during seed germination.

Cultivated peanut (2n = 4x = 40, AABB) is an allotetraploid derived from wild diploid peanut, Arachis duranensis (AA) and Arachis ipaensis (BB), and its A and B subgenomes showed 0.88% and 12.46% expansion in gene content compared with its progenitors during polyploid evolution (Bertioli et al., 2019). Gene duplications are considered to be one of the primary driving forces in the evolution of genomes and gene families and reportedly account for 8~20% of the genes in eukaryotic genomes (Bowers et al., 2003; Moore & Purugganan, 2003). The results of our collinearity analysis showed that the 58 members of the AhXTH family mostly come from tandem duplications and segmental duplications and that purifying selection on codons is the dominant mode during gene amplification, indicating that most XTH genes were highly conserved during the evolutionary process. Only two gene pairs, AhXTH1/AhXTH27 and AhXTH12/AhXTH42, both with segmental duplication, underwent positive selection, likely resulting from environmental adaptation. Tandem duplication resulted in four gene clusters in our study, all of which belonged to Group I/II. Here, AhXTH3 and 34, 4 and 33, 5 and 32, 6 and 29, 21 and 51, and 22 and 52, respectively, were found to be orthologous genes that might have existed before allotetraploidy occurred. The gene order of the clusters on chromosomes 01/11 (AhXTH3, 4, 5 and 6 on chr. 01, and AhXTH34, 33, 32 and 29 on chr. 11) were found in the opposite direction, which might be due to the presence of an inversion on chr. 01/11 (Bertioli et al., 2019). Our results also showed that AhXTH genes were unevenly distributed in the A and B genomes; there were six more AhXTHs on chromosomes 11–20 than on chromosomes 1–10. Cannon et al. (2004) indicated that tandem duplication often results from unequal crossing-over and is an important engine producing new gene copies in genomic clusters. The cluster members AhXTH30 and 31 on chr. 11, having lower sequence similarity compared to the other genes in the same cluster, might have evolved by the crossover between homologous chromosome segments.

In general, the conservation of the protein sequence and structure determines the conservation of its function, and some structural differences might lead to divergence in enzyme activity or function. We found that the AhXTH protein structures in three different subfamilies were conserved: they all contained a highly conserved ExDxEx, that is, the region of the enzyme active site, and the sequences of their loop3s were relatively conserved. However, the sequences of loop2 and the N-glycosylation site, and the length of their interval, differed between Group I/II and Group III. Previous research indicated that all members of the XTH family exhibit xyloglucan endo-transglucosylase (XET) activity, and only Group IIIA has a combined function of XET and xyloglucan endo-hydrolase (XEH) (Baumann et al., 2007). Arabidopsis XTH31, a member of Group IIIA, has been confirmed to have high XEH activity and low XET activity in vitro (Zhu et al., 2012). Additionally, the variation in the length of loop2 was a prerequisite for distinguishing the activity of XET and xyloglucanase but was not sufficient (Baumann et al., 2007). Indeed, the truncated loop2 of TmNXG1 was demonstrated to result in an increase in transglycosylation ability and a decease in hydrolysis ability (Baumann et al., 2007). Our results showed that the loop2 of AhXTH in Group IIIA extends by three to four residues at the C-terminus compared to that in other groups, indicating that these enzymes might have a higher transglycosylation:hydrolysis ratio. Although there were consistent lengths for loop2 in Group IIIB and Group I/II, the amino acid compositions were different between these two groups. In addition, sequence alignment showed that the extended loop1 in Group IIIB was similar to that in TmNXG1. Xu et al. (2010) found that compared to TmXTH1, an enzyme with XEH activity, the members of subfamily IIIB from Populus trichocarpa have a truncated loop2, but other structural characteristics might result in a high possibility of functioning as XEH instead of XET. Thus, whether the IIIB proteins of the peanut XTH family have XEH functions needs to be investigated further.

Multiple studies have shown that some XTH genes play a vital role in seed germination and act on the xyloglucan chain in the cell wall, thus promoting cell wall expansion and repair (Dogra, Sharma & Yelam, 2016; Hernandez-Nistal et al., 2006; Nonogaki, 2019; Sangi et al., 2019; Sechet et al., 2016; Tomomi et al., 2004). Xyloglucan, as a kind of reserve, could also be hydrolyzed and mobilized by XTH to provide energy for the germination of seeds (Campbell & Braam, 1999). However, different XTHs may have unique temporal and spatial expression patterns and exert distinct biological functions. The tomato LeXTH4 gene is specifically expressed in the micropylar endosperm cap of germinating seeds prior to radicle emergence and plays a role in weakening the endosperm (Chen, Nonogaki & Bradford, 2002). In chickpeas, CaXTH1 is involved in the elongation of epicotyls and embryonic axes during seed germination. The expression of CaXTH1 was detectable as early as 1 h after seed imbibition and reached a peak at 24 h, after which the epicotyl began to develop (Hernandez-Nistal et al., 2006; Romo et al., 2005). Arabidopsis AtXTH31/XTR8 was associated with reinforcing the cell wall of the endosperm and slowing seed germination. In this study, we found that during peanut seed germination, many AhXTHs were significantly upregulated, with the majority belonging to Groups I/II and IIIB. These genes might play key roles in seed germination under different regulatory mechanisms. Evolutionarily, orthologous genes from different species may perform similar functions. Therefore, some results of previous studies could provide clues for further characterization of XTH functions in peanut. In this study, we verified that AhXTH4 might be involved in the cell expansion of hypocotyls during seed germination and the growth of etiolated seedlings based on phenotypic data from a complementary experiment on Arabidopsis xth22 transgenic lines and the expression profile of AhXTH4. Significantly, some peanut XTH genes with higher homology had obviously divergent expression patterns at the FS, DS and GS stages in the present research. Allotetraploid peanuts have undergone parallel evolution and polyploidization of the wild diploid genome, resulting in heterogeneous expression patterns or neofunctionalization of some homologous genes (Wang et al., 2019). These divergences in expression patterns and the functional diversity of family members are the results of long-term evolution caused by adaptation to environmental stress (Becnel et al., 2006; Rose et al., 2002; Xu et al., 1995; Yokoyama & Nishitani, 2001).

Hormones control the growth and development of plants by regulating the expression of various genes. As an important part of the plant life cycle, the process of seed germination is also synergistically regulated by various hormones (Gazzarrini & Tsai, 2015; Holdsworth, Bentsink & Soppe, 2008; Kucera, Cohn & Leubner-Metzger, 2005; Penfield, 2017). Several XTH genes in Arabidopsis displayed positive or negative responses to plant hormones, including auxin, BR, GA and abscisic acid (ABA), during plant growth and development (Yokoyama & Nishitani, 2001). The expression of the CaXTH1 gene was specifically upregulated in hypocotyls of chickpea under the induction of indole-3-acetic acid (IAA) or BR, and at the same time, its transcripts showed their highest level when IAA plus BR was applied during seed germination (Romo et al., 2005). The accumulation of LeXET2 in the stem and hypocotyl was significantly enhanced under GA induction (Catala et al., 2001). Previous studies found that AtXTH31/XTR8 was expressed in an endosperm-specific pattern, and its expression was induced by salicylic acid (SA) (Miura et al., 2010). Furthermore, it was found that the mutation of this gene caused xtr8 to be less sensitive to ABA and to germinate faster (Endo et al., 2012). Here, analysis of the regulatory regions of AhXTHs showed that many phytohormone-responsive elements, including EREs, ABREs, gibberellin-responsive elements (GAREs), TGA elements and AuxRR core motifs involved in auxin responsiveness, exist in these regions. Our previous analysis also indicated that the coordination of multiple hormone signaling plays a crucial role during seed germination in peanut (Xu et al., 2020). Therefore, it was speculated that some AhXTHs upregulated in GS might respond to hormones and control the process of radicle protrusion and seed germination. However, the functions of these AhXTH genes need further investigation.

Conclusions

In this study, 58 members of the AhXTH family were identified and divided into Groups I/II, IIIA and IIIB in cultivated peanut. All AhXTH genes were scattered on 18 chromosomes, with the exception of chr. 07 and 17, and they had relatively conserved exon-intron patterns mostly with three to four introns. The AhXTH family exhibited many replication events, including 42 pairs of segmental duplications and 23 pairs of tandem duplications, during genome evolution. Their encoded proteins contained the conserved ExDxEx domain and N-linked glycosylation sites and displayed conserved secondary structures (loops1–3) in members of the same group. The relative expression levels of 45% of family genes were upregulated during seed germination, implying the important roles of AhXTHs in regulating seed germination. The roles of AhXTH4 in the cell expansion of the hypocotyl during seed germination and the growth of etiolated seedlings were verified by the complementary phenotype of Arabidopsis xth22 based on its overexpression lines and the expression profile of AhXTH4. The biological functions of other genes require verification with further experiments.

Supplemental Information

Supplemental Information 1 qRT-PCR primers for AhXTHs and reference genes.

Click here for additional data file.

Supplemental Information 2 Primers of Arabidopsis col-0 and mutant xth22.

Click here for additional data file.

Supplemental Information 3 Coding sequences of Peanut AhXTHs.

Click here for additional data file.

Supplemental Information 4 Protein sequences of Peanut AhXTHs.

Click here for additional data file.

Supplemental Information 5 The multiple alignment of amino acid sequences among 58 AhXTHs in peanut and 1UN1.

Amino acid sequences from 58 AhXTHs in peanut were aligned using PttXET16-34 (1UN1) as a referent sequence by MEGA X, and their secondary structures were predicted using ESPript. The conserved residues were shown in blue frames.

Click here for additional data file.

Supplemental Information 6 The multiple alignment of amino acid sequences among 58 AhXTHs in peanut and 2UWA.

Amino acid sequences from 58 AhXTHs in peanut were aligned using TmNXG1(2UWA) as a referent sequence by MEGA X, and their secondary structures were predicted using ESPript. The conserved residues were shown in blue frames.

Click here for additional data file.

Supplemental Information 7 The FPKM value of AhXTH genes at DS, FS and GS stages.

Click here for additional data file.

Supplemental Information 8 Pearson correlation coefficient between RNA-seq and qRT-PCR.

Q.FS, Q.DS, and Q.GS respectively denote the qRT-PCR results in the freshly harvested seeds (FS), the dried seeds (DS) and the newly germinated seeds (GS). R.FS, R.DS, and R.GS indicate the analysis results of RNA-seq data in FS, DS and GS, respectively.

Click here for additional data file.

Supplemental Information 9 The raw data of Ct value used for qRT-PCR in Fig. 5.

Click here for additional data file.

Supplemental Information 10 Melting curve analysis of qRT-PCR.

Melting curve analysis was performed during the qRT-PCR precedure. The letter a-n respectively indicate the analysis results of AhXTH4, AhXTH5, AhXTH14, AhXTH15, AhXTH16, AhXTH24, AhXTH30, AhXTH31, AhXTH35, AhXTH38, AhXTH42, AhXTH52, ACTIN and UBI.

Click here for additional data file.

Supplemental Information 11 Agar gel electrophoresis of PCR products of AhXTH promoters.

2K: DNA molecular marker (TaKaRa DL2000); 1 and 2: PCR products of the promoters of AhXTH4 and AhXTH22

Click here for additional data file.

Additional Information and Declarations

Competing Interests

Author Contributions

Data Availability

The authors declare that they have no competing interests.

Jieqiong Zhu performed the experiments, analyzed the data, prepared figures and/or tables, authored or reviewed drafts of the paper, and approved the final draft.

Guiying Tang performed the experiments, authored or reviewed drafts of the paper, and approved the final draft.

Pingli Xu performed the experiments, authored or reviewed drafts of the paper, and approved the final draft.

Guowei Li analyzed the data, authored or reviewed drafts of the paper, and approved the final draft.

Changle Ma analyzed the data, authored or reviewed drafts of the paper, and approved the final draft.

Pengxiang Li performed the experiments, prepared figures and/or tables, authored or reviewed drafts of the paper, and approved the final draft.

Chunyu Jiang performed the experiments, analyzed the data, authored or reviewed drafts of the paper, and approved the final draft.

Lei Shan conceived and designed the experiments, analyzed the data, authored or reviewed drafts of the paper, and approved the final draft.

Shubo Wan analyzed the data, authored or reviewed drafts of the paper, and approved the final draft.

The following information was supplied regarding data availability:

The raw measurements are available in the Supplemental Files.

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
