# Peer review of "Genome-wide identification of xyloglucan endotransglucosylase/hydrolase gene family members in peanut and their expression profiles during seed germination"

_PeerJ, doi:10.7717/peerj.13428_

## Round 0.1 · original submission · Major Revisions

The reviewers have provided some useful comments and suggestions to improve the manuscript. Many of the comments have to do with the interpretation of data and should be relatively easy to address.

Reviewer 1 ·

Basic reporting

In this paper, Zhu and colleagues report a genome-wide survey of the Xyloglucan Endotransglucosylase/Hydrolase (XTH) gene family in peanut. The authors performed phylogenetic analysis, compared exon-intron structures, presented genomic distribution, conserved motifs, and analyzed gene expression profiles using publicly available transcriptome data of XTH family genes in peanut. Based on these analyses, the authors found that segmental and tandem duplications contributed significantly to the evolution of XTH proteins in peanut. Moreover, they identified a subset of XTH gene that might function in the seed germination process.
Although the subject of the manuscript fits well with the scope of PeerJ, the manuscript suffers from substantial weakness in several aspects in data analysis. Therefore, it must be subjected to major compulsory revisions prior to considering for possible publication.

Experimental design

The subject of the manuscript fits with the aim and scope of the PeerJ.

Validity of the findings

The analysis is generally sound and acceptable. Conclusions were drawn from the analysis.

Additional comments

Major revisions:
1. Only two species i.e. Arabidopsis and soybean were included in the phylogenetic analysis. Given that numerous plant genomes have already been sequenced to date, the authors are recommend to include several more representative species to clearly elucidate the evolutionary relationships among XTH genes in plants.
2. The authors only described how to define segmental duplication in the Materials and Methods section. However, they did not describe how to define the tandem duplication? It should be clarified what criteria were adopted to justify a tandem duplication.
3. The authors showed that one XTH gene usually had more than one paralogous genes originated from the segmental duplication. For example, AhXTH8 has three different paralogous pairs namely AhXTH20, AhXTH37, and AhXTH50 derived from segmental duplication. However, in the phylogenetic tree (Figure 3), it showed that AhXTH8 is most closely associated with AhXTH37. How to interpret these results? Is it consistent with the chromosomal duplication event during the evolution?
4. The authors detected several clustered XTH gene on the chromosome. They defined them into several paralogous pairs derived from tandem duplication. For example, they defined 5 paralogous pairs i.e. AhXTH3 vs AhXTH4, AhXTH3 vs AhXTH6, AhXTH3 vs AhXTH5, AhXTH4 vs AhXTH6, AhXTH5 vs AhXTH6, for one cluster of XTHs. The same situation also applied to AhXTH29-AhXTH34. Since no criteria were described in defining the tandem duplication as I have mentioned above, these analysis seems questionable.
5. The authors predicted the regulatory elements of hormone signaling in the promoter region of AhXTH genes (Supplementary Figure 1). These results should not be presented in the Discussion section, it should be presented in the Results section wherever is appropriate.
6. To verify their conclusions drawn from the expression analysis and clarify its functional roles, the authors should selected at least one gene to study its functional roles by transforming into Arabidopsis null mutant and/or wild type.
7. The language of the manuscript needs substantial improvement as there are numerous improper descriptions throughout the manuscript. In addition, the manuscript should be critically revised by a native English speaker to eliminate the grammar errs and improper wording.

Minor revisions:
1. No scale bar was provided for chromosome length in Figure 1
2. Segmental duplication was most frequently used throughout the manuscript. However, fragment duplication was occasionally used, it should be unified in description
3. Subfamily I/II was not marked in Figure 3. Bootstraps lower than 50 was not recommend to display at the branch.
4. The gene names in Figure 5 should be italic. Genes with no expression values at FS, DS and GS stages should be excluded in the analysis.

·

Basic reporting

The manuscript meets most of the basic reporting.

Experimental design

Experimental design well planned and executed.

Validity of the findings

Findings are valid. However, some comments are to be addressed.

Additional comments

• Can qRT-PCR for a few specific genes be considered as transcriptome analysis?
• Clearly mention how many genes were considered for expression analysis.
• Was there any effort made to check for primer efficiency and melting curve analysis?
• Structural analysis of the homologous AhXTH genes has been partly presented. More importantly, the sequence similarity among the homologs can be indicated along with the relevance.
• More analysis on the function of AhXTH genes would be useful. The DNA tagged mutants of Arabidopsis can be considered to gain more knowledge on the function.
• Care should be taken to italicize the scientific names and maintain consistency in the main body.

---

## Round 0.2 · Minor Revisions

We have received a second round of reviews for you manuscript. Most of the requested changes are aimed at improving the clarity of the manuscript or editorial changes to improve the writing, Please address the issues raised by the reviewers and make additional edits to improve the writing. Reviewer 2 has highlighted some especially problematic lines.

Reviewer 1 ·

Basic reporting

The authors have responded to all of my concerns raised in last review. Now, the manuscript can be accepted pending minor revisions as follows:
1. There are still some typos and format errors in the manuscript. For instance, Circos instead of Circus, bar instead of ber etc; gene names in Figure 5 should be italic; the naming of promoter:GUS transgenic lines in Figure 7 should be the same as in the text. Pod should be silique for Arabidopsis.
2. The descriptions of figure legend for Figure 7 were not provided in detail. What does the positive control refer to ?
3. The hypocotyl length of Col-0 and AhXTH4 complemented lines in Figure 8 should better be quantitatively measured and present in a Bar graph.

Experimental design

no comment

Validity of the findings

no comment

Additional comments

no comment

·

Basic reporting

This study reports genome-wide identification of xyloglucan endotransglucosylase/hydrolase gene family members, their structural and functional characterization and expression profiles during seed germination in peanut. In total, 58 AhXTH genes were identified.
Line # 146: "If two sequences shared over 70% identity and covered over 70% of their sequences, then they were considered to be homologous genes and to have collinearity.". Please provide a reference.
Naming of the members of XTH genes can be revised to make them uniform and clear.
It is interesting to know their distribution on A and B genomes and the extent of homeologous nature of these genes. Of them, 26 AhXTH genes high-level expression during seed germination. Probable function of the remaining 32 AhXTH genes may be provided in the Results.
Promoter analysis and ectopic expression analysis indicated the functional roles of the selected AhXTH genes.
Line # 197: Promoter of 2000 bp was isolated by PCR. Kindly give the details on the PCR. Also tell about delimitting the promoter region.
The details on the RT-PCR like PCR efficiency, melting curve analysis, fold changes etc may be useful for the readers.
Kindly provide the collinearity between the results of RNA-Seq and RT-PCR.
The English language should be improved to ensure that an international audience can clearly understand your text. Some examples where the language could be improved include Line # 40, 113, 115-116, 127, 226 etc

Experimental design

See the above comment

Validity of the findings

See the above comment

Additional comments

See the above comment

---

## Round 0.3 · accepted · Accept

Thank you for addressing the comments raised by the reviewers. I am satisfied that their concerns have been appropriately resolved in your revised manuscript.